# ADAPTIVE LOCAL TRAINING IN FEDERATED LEARNING

**Donald Shenaj**
University of Padova

**Eugene Belilovsky**
Mila, Concordia University

**Pietro Zanuttigh**
University of Padova

## ABSTRACT

Federated learning is a machine learning paradigm where multiple clients collaboratively train a global model by exchanging their locally trained model weights instead of raw data. In the standard setting, every client trains the local model for the same number of epochs. We introduce ALT (Adaptive Local Training), a simple yet effective feedback mechanism that can be exploited at the client side to limit unnecessary and degrading computations. ALT dynamically adjusts the number of training epochs for each client based on the similarity between their local representations and the global one, ensuring that well-aligned clients can train longer without experiencing client drift. We evaluated ALT on federated partitions of the CIFAR-10 and Tiny-ImageNet datasets, demonstrating its effectiveness in improving model convergence and stability.

## 1 INTRODUCTION

Federated learning (FL) (McMahan et al., 2017) has emerged as a machine learning approach prioritizing privacy while fostering collaborative training, avoiding centralized data storage concerns.

Typically, in FL every client performs training for the same number of local epochs (Li et al., 2021; Shenaj et al., 2023). However, different clients could have different computational capabilities, and communication speeds, and the server might request updates when the training is not yet concluded. Moreover, training each client for a fixed pre-defined number of steps could lead to client drift.

For this reason, we train for a variable number of local epochs across clients and training rounds, similarly to (Li et al., 2020; Michieli et al., 2022). In particular, we study the effect of dynamic local epochs at the client side and propose a client-side control strategy to mitigate representation drift, reduce communication costs, and improve performances.

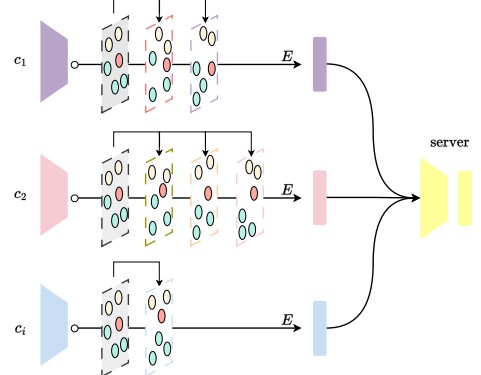

Figure 1: Overview of the proposed federated learning strategy with dynamic local epochs.

**Related Work.** FedProx (Li et al., 2020) introduces a framework that allows clients to perform variable amounts of local training, assuming that some may not complete their updates within a fixed time window. It also employs a regularizer to mitigate the impact of inconsistent updates.

Similarly, SCAFFOLD (Karimireddy et al., 2020) tackles client drift by estimating and correcting update directions for both the server and clients, ensuring more stable local updates. However, while these methods offer improvements, they fail to achieve significant performance gains over FedAvg in deep neural networks training, particularly when applied in realistic computer vision applications. To address this limitation, MOON (Li et al., 2021) proposes a model-based contrastive learning approach to enhance local training in deep networks by enforcing similarity between the local model and the incoming global one, while discouraging similarity with the previous local model.

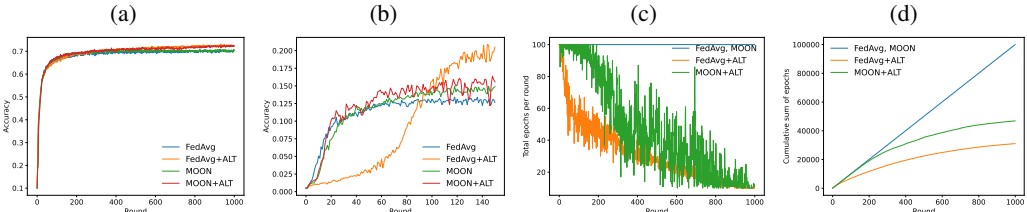

Figure 2: (a) Accuracy curve on CIFAR-10. (b) Accuracy curve on Tiny-ImageNet. (c) Total number of epochs per round on CIFAR-10. (d) Cumulative epochs per round on CIFAR-10.

Finally, Dap-FL (Chen et al., 2023) enables resource-constrained participants to adaptively adjust local training hyper-parameters including the number of epochs.

Building on these insights, we introduce a simple yet effective mechanism to control the length of local training, opening new research opportunities in adaptive and dynamic federated learning. An overview of our setting is shown in Fig. 1.

## 2 METHOD

Let us assume a set of clients $\mathcal{K}$, where each client $k \in \mathcal{K}$ has access to a local set of samples $(\mathcal{X}_k, \mathcal{Y}_k)$ from a dataset $\mathcal{D}_k$ with $|\mathcal{D}_k| = n_k$. At each communication round $r \in \{1, \ldots, R\}$, the server selects a subset of clients $S \subset \mathcal{K}$ and sets an adaptive threshold $T_h(r) = a + \frac{b \cdot r}{R}$, that linearly increases during the training (we set $a = 0.1$ and $b = 0.8$, see Appendix B for more details). Each client $s \in S$ then initializes its local model $\theta_s^r$ with the global model $\theta^r$ and trains for $E_s^r$ local epochs, where $E_s^r$ varies across clients and rounds.

At each training step, as usual in neural networks training, the local model is updated using gradient descent on mini-batches $\mathcal{B} \subset \mathcal{D}_s$, i.e., $\theta_s^r \leftarrow \theta_s^r - \eta \nabla \mathcal{L}_s(\theta_s^r; \mathcal{B})$.

Let us denote with $p_s = f(w_s, \mathcal{B})$ and $p_g = f(w_g, \mathcal{B})$ the feature embeddings of the local and global models for the considered batch: the local training halts as soon as the similarity condition $\cos(p_s, p_g) < T_h(r)$ is met, i.e., when the difference between the embeddings is smaller than the threshold. If the condition is never met, it will stop as usual after the maximum number of local epochs $E$ is reached. Note that the threshold increases with time, i.e., the criteria becomes more and more strict while the difference w.r.t. the starting model typically increases.

After local training, the client models are sent to the server, which aggregates them using standard federated averaging, i.e.,: $\theta^{r+1} = \sum_{s \in S} \frac{n_s}{n} \theta_s^r$. The process is repeated for R rounds. The algorithm is detailed in Appendix C.

## 3 RESULTS

We evaluate the performance of our algorithm on the CIFAR-10 (Krizhevsky & Hinton, 2009), and Tiny-ImageNet (Le & Yang, 2015) datasets. As a strong baseline for local training, we consider MOON. We consider $|\mathcal{K}| = 100$ clients (with $10\%$ participation), and the data is partitioned according to the Dirichlet distribution with the concentration parameter $\alpha = 100$. By looking at Figure 2, we can notice that our method allows to reduce substantially carbon footprint by reducing the cumulative epochs per round (sum of all clients epochs) (Fig. 2c and 2d), and leads also to improved performance (Fig. 2a and 2b). In particular, we notice that FedAvg + ALT (FedALT) could work similarly or better than MOON, while being much more efficient.

## 4 CONCLUSION

In this work, we introduced a representation learning feedback mechanism to control the number of local epochs reducing energy consumption and communication costs which can be seamlessly integrated into FL algorithms.

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

## APPENDIX

## A IMPLEMENTATION DETAILS

We adopt as feature extractor $f$ a simple CNN with 2 convolutional layers for CIFAR-10, and ResNet-50 for Tiny-ImageNet. We use the SGD optimizer with a learning rate of 0.01 for all approaches. The SGD weight decay is set to 0.00001 and SGD momentum is set to 0.9. The batch size is set to 64. The maximum number of local epochs $E$ is set to 10, and the number of communication rounds R is 1000 for CIFAR-10 and 150 for Tiny-ImageNet.

## B SELECTION OF THE THRESHOLD FUNCTION

We experimented different threshold functions $T_h(r)$, including 3 simple approaches:

- Linear Increasing: $T_h(r) = 0.1 + 0.8 \cdot \frac{r}{R}$
- Linear Decreasing: $T_h(r) = 0.9 - 0.8 \cdot \frac{r}{R}$
- Fixed: $T_h(r) = c$, where $c$ is a constant.

Among these strategies, the most effective was the linear increasing threshold. This approach enables a 'slow start,' where early training rounds allow for more flexibility in local updates before gradually enforcing stricter similarity constraints. This progressive tightening helps balance exploration and convergence, leading to improved overall model performance. However, in practice, any thresholding (including a fixed rule) is useful to reduce carbon footprint.

## C   ALGORITHM

1: **Input:** Initialize model parameters $\theta$
2:          Initialize maximum rounds $R$
3:          Initialize threshold parameters a, b
4: **Server executes:**
5: **for** $r = 1$ **to** $R$ **do**
6:      $S \leftarrow$ Random subset of clients
7:      $T_h(r) \leftarrow a + \frac{b*r}{R}$
8:      **for** each client $i \in S$ **do**
9:          $\theta_i \leftarrow$ ClientUpdate$(i, \theta, r, T_h(r))$
10:      **end for**
11:      $\theta \leftarrow \sum_{i \in S} \frac{n_i}{n} \theta_i$
12: **end for**
13: **ClientUpdate**$(i, \theta, r, T_h(r))$**:**
14: **if** $r = 1$:
15:      $\theta_i \leftarrow \theta$
16:      $\theta_i := \{w_i, v_i\}$
17: $\theta_g \leftarrow \theta$
18: $\theta_g := \{w_g, v_g\}$
19: stop $\leftarrow False$
20: **for** $j = 1, 2, ..., E$ and stop $= False$ **do**
21:      **for each** batch $\mathcal{B}$ in $\mathcal{D}_i$ **do**
22:          $p_i \leftarrow f(w_i, \mathcal{B})$
23:          $p_g \leftarrow f(w_g, \mathcal{B})$
24:          **if** $\cos(p_i, p_g) < T_h(r)$: stop $\leftarrow True$
25:          $\theta_g \leftarrow \theta_g - \eta \nabla \mathcal{L}(\theta_g; \mathcal{B})$
26: $\theta_i \leftarrow \theta_g$
27: **return** $\theta_i$

**Algorithm 1:** Implementation of FedAvg with the ALT stopping criteria (FedALT)

