# OpenReview forum: "Adaptive Local Training in Federated Learning"
_ICLR.cc/2025/Workshop/MCDC — MCDC @ ICLR 2025_

### Official Review · Reviewer_Rakc · 2025-02-22

**Rating:** 8
**Confidence:** 3
**Fit:** 4

**Summary:**

This paper presents Adaptive Local Training (ALT), a novel method of dynamically adjusting training epochs per client based on cosine similarity measures of client model's embeddings and global model's embeddings. This serves to avoid excessive computation and mitigate client drift in cases of heterogenous data distribution. The authors demonstrate improved performance and reduced computational costs on CIFAR-10 and TinyImageNet datasets compared to baselines like FedAvg and MOON.

**Reason For Giving A Higher Score:**

1. Novel and straightforward approach with a detailed method.
2. Well presented paper
3. Sufficient experiments

**Reason For Giving A Lower Score:**

1. Limited theoretical analysis.

**Strengths And Weaknesses:**

Strengths:

1. Novel yet straightforward proposal to reduce computational load and excessive training.
2. Well-written paper with a clear motivation.
3. Decent generalization shown across datasets and comparison with baseline methods.

Weaknesses:

1. Results not presented with different threshold functions.
2. Limited theoretical analysis of why the method works.

**Suggestions:**

1. Consider adding ablation study with threshold functions.
2. Provide theoretical explanations of why exactly the method works.
3. Consider referencing Chen et al. "Dap-FL: Federated Learning Flourishes by Adaptive Tuning and Secure Aggregation".

---

### Official Review · Reviewer_q5KB · 2025-02-26

**Rating:** 7
**Confidence:** 4
**Fit:** 4

**Summary:**

This paper proposed an Adaptive Local Training framework that dynamically adjusts the number of local training epochs for each client, addressing challenges like client drift in heterogeneous data settings. Empirical results on CIFAR-10 and TinyImageNet show the effectiveness of the proposed method.

**Reason For Giving A Higher Score:**

The paper lacks a convergence analysis of the proposed method, and the experimental validation is insufficient.

**Reason For Giving A Lower Score:**

The proposed method offers new insights for addressing client drift and data heterogeneity.

**Strengths And Weaknesses:**

## Strengths
The proposed ALT framework provides new ideas and methods for addressing client drift and data heterogeneity by dynamically adjusting the number of local training epochs based on the similarity between local and global model representations. Moreover, the proposed method is simple and effective and can be integrated into existing methods like FedAvg.

## Weaknesses
1. The paper lacks relevant theoretical analysis.
2. The paper lacks additional experiments to discuss the impact of varying degrees of data heterogeneity on the performance of the method.

**Suggestions:**

See weaknesses. Additionally, more baseline methods could be included to demonstrate the flexibility and adaptability of the proposed ALT.

---

### Official Review · Reviewer_nUYo · 2025-02-27

**Rating:** 3
**Confidence:** 3
**Fit:** 4

**Summary:**

This paper introduces Adaptive Local Training (ALT) in Federated Learning (FL), a novel method that dynamically adjusts the number of training epochs for each client based on the similarity between local and global model representations.
In traditional FL, all clients train for a fixed number of epochs per round, regardless of how closely their models align with the global model. In contrast, ALT halts training early for clients whose local representations have sufficiently aligned with the global model. More concretely, for each client kk, training stops when the cosine similarity:

\[
\cos(\mathbf{p}_s, mathbf{p}_g) < Th(r).
\]
 Falls below an adaptive threshold

\[
Th(r) = a + b \frac{r}{R}
\]

where \(r\) is the current round and \(R\) is the total number of rounds.

The paper evaluates ALT on CIFAR-10 and Tiny ImageNet using its integration into FedAvg and MOON, demonstrating that it reduces total training epochs while maintaining comparable or improved accuracy.

**Reason For Giving A Higher Score:**

ALT provides a simple mechanism for reducing computation in FL by dynamically adjusting client training epochs. Its strengths include:

- Reduced computational overhead while maintaining accuracy.
- Compatibility with existing FL algorithms (FedAvg, MOON).
- Clear empirical benefits demonstrated on CIFAR-10 and Tiny ImageNet.

**Reason For Giving A Lower Score:**

The paper suffers from several major weaknesses that limit its impact to the field:

- Unclear novelty - adaptive training in FL has been explored before, and ALT does not introduce a fundamentally new concept.
- Lack of rigorous theoretical grounding - it is unclear to me what the theoretical rationale is behind the formulation of the linear schedule.
- Weak experimental baselines - a simple method that gradually reduces epochs per round was not tested, leaving uncertainty about whether ALT’s benefits can be similarly met by using a non-adaptive approach that simply reduces the number of epochs per round
- Training instability - the curve (Figure b) contradicts claims of improved convergence. The gradient of the curve is still markedly positive at the limit of the graph and has not converged
- Superficial conclusions - the results discussion is weak, with a tenuous claim relating to implications for energy efficiency that is not well-supported by the data.

Without addressing these issues, the paper falls short of making a significant contribution to the field.

**Strengths And Weaknesses:**

Strengths:
Key concepts are supported by figures - (a), (c), and (d) illustrate the effectiveness of ALT in reducing total epochs while maintaining accuracy.
Easy integration with other FL algorithms - ALT can be used with existing methods like FedAvg and MOON
Demonstrates improvements in computational efficiency without training degradation

Weaknesses:
- Training instability - figure (b) shows a sinusoidal training curve that has clearly not converged at the limit of the graph, with no evidence therefore included in the paper that training is robust on the Tiny ImageNet dataset.
- Lack of clear causality - it is unclear whether ALT’s improvements stem from the adaptive schedule or simply from reducing the number of training epochs per round. A simple baseline that uniformly decreases the number of epochs per round should have been tested.
- Weak conclusion - The conclusions restate general benefits (e.g., energy savings) but do not effectively summarize key empirical results.
- Incorrect figure labelling - figure (c) is mislabeled—the y-axis should be "Total Epochs Per Round" instead of cumulative epochs.
- Questionable novelty - adaptive local training in FL has been explored in FedProx (Li et al., 2020) and SCAFFOLD (Karimireddy et al., 2020), though ALT introduces a specific similarity-based criterion
- Lack of theoretical justification - there is no explanation for the choice of the adaptive threshold equation beyond empirical results.
- No figure references in the text - figures are not referenced within the main paper, making it harder to connect findings to visual results.
- Weak justification for parameter selection - the values of \( a \) and \( b \)  are not empirically justified, nor is there an ablation study exploring different settings
- Limited non-i.i.d. evaluation - the single concentration parameter \alpha = 100 results in a data split that is almost i.i.d. A lower \alpha should have been tested to assess performance under stronger non-i.i.d. partitions.
- Throwaway statements included throughout the paper that are non-specific - for example, “In addition to that, when the clients’ data distribution is very heterogeneous, training each client for a fixed pre-defined number of steps leads to client drift and complicates the aggregation.” What does complicating the aggregation mean?

**Suggestions:**

- Reference figures within the paper
- Not clear whether improvement is related to the schedule. You might see similar performances if you simply dropped the number of epochs each client trains for each round. This wasn’t tested
- Rationale as to the choice of parameters a and b - an empirical analysis would be sufficient
- Test a variety of different values for the concentration parameter \alpha to see whether the method can handle non-i.i.d. is sensitive to partitions which are more different.

---

### Decision · Program_Chairs · 2025-03-06

**Decision:**

Accept

**Comment:**

This paper proposed an Adaptive Local Training framework for clients in an FL setup that dynamically adjusts the number of local training epochs for each client, addressing challenges like client drift in heterogeneous data settings. The paper received scores with high variance. We suggest the authors incorporate the comments and suggestions from reviewer nUYo to strengthen the paper. The paper seems relevant to the topic of decentralized training. Overall, we recommend accepting this work to the workshop.